# Attention Based Joint Learning for Supervised Electrocardiogram Arrhythmia Differentiation with Unsupervised Abnormal Beat Segmentation

## Abstract

Deep learning has shown great promise in arrhythmia classification in electrocardiogram (ECG). Existing works, when classifying an ECG segment with multiple beats, do not identify the locations of the anomalies, which reduces clinical interpretability. On the other hand, segmenting abnormal beats by deep learning usually requires annotation for a large number of regular and irregular beats, which can be laborious, sometimes even challenging, with strong inter-observer variability between experts. In this work, we propose a method capable of not only differentiating arrhythmia but also segmenting the associated abnormal beats in the ECG segment. The only annotation used in the training is the type of abnormal beats and no segmentation labels are needed. Imitating human's perception of an ECG signal, the framework consists of a segmenter and classifier. The segmenter outputs an attention map, which aims to highlight the abnormal sections in the ECG by element-wise modulation. Afterwards, the signals are sent to a classifier for arrhythmia differentiation. Though the training data is only labeled to supervise the classifier, the segmenter and the classifier are trained in an end-to-end manner so that optimizing classification performance also adjusts how the abnormal beats are segmented. Validation of our method is conducted on two dataset. We observe that involving the unsupervised segmentation in fact boosts the classification performance. Meanwhile, a grade study performed by experts suggests that the segmenter also achieves satisfactory quality in identifying abnormal beats, which significantly enhances the interpretability of the classification results.

## 1 Introduction

Arrhythmia in electrocardiogram (ECG) is a reflection of heart conduction abnormality and occurs randomly among normal beats. Deep learning based methods have demonstrated strong power in classifying different types of arrhythmia. There are plenty of works on classifying a single beat, involving convolutional neural networks (CNN) (Acharya et al., 2017b; Zubair et al., 2016), long short-term memory (LSTM) (Yildirim, 2018), and generative adversarial networks (GAN) (Golany & Radinsky, 2019). For these methods to work in clinical setting, however, a good segmenter is needed to accurately extract a single beat from an ECG segment, which may be hard when abnormal beats are present. Alternatively, other works (Acharya et al., 2017a; Hannun et al., 2019) try to directly identify the genres of arrhythmia present in an ECG segment. The limitation of these works is that they work as a black-box and fail to provide cardiologists with any clue on how the prediction is made such as the location of the associated abnormal beats.

In terms of ECG segmentation, there are different tasks such as segmenting ECG records into beats or into P wave, QRS complexity, and T wave. On one hand, some existing works take advantage of signal processing techniques to locate some fiducial points of PQRST complex so that the ECG signals can be divided. For example, Pan-Tompkins algorithm (Pan & Tompkins, 1985) uses a combination of filters, squaring, and moving window integration to detect QRS complexity. The shortcomings of these methods are that handcraft selection of filter parameters and threshold is

needed. More importantly, they are unable to distinguish abnormal heartbeats from normal ones. To address these issues, Moskalenko et al. (2019); Oh et al. (2019) deploy CNNs for automatic beat segmentation. However, the quality of these methods highly depends on the labels for fiducial points of ECG signals, the annotation process of which can be laborious and sometimes very hard. Besides, due to the high morphological variation of arrhythmia, strong variations exist even between annotations from experienced cardiologists. As such, unsupervised learning based approaches might be a better choice.

Inspired by human's perception of ECG signals, our proposed framework firstly locates the abnormal beats in an ECG segment in the form of attention map and then does abnormal beats classification by focusing on these abnormal beats. Thus, the framework not only differentiates arrhythmia types but also identifies the location of the associated abnormal beats for better interpretability of the result. It is worth noting that, in our workflow, we only make use of annotation for the type of abnormality in each ECG segment without abnormal beat localization information during training, given the difficulty and tedious effort in obtaining the latter.

We validate our methods on two datasets from different sources. The first one contains 508 12-lead ECG records of Premature Ventricular Contraction patients, which are categorized into different classes by the origin of premature contraction (e.g., left ventricle (LV) or right ventricle (RV)). For the other dataset, we process signals in the MIT-BIH Arrhythmia dataset into segments of standard length. This dataset includes various types of abnormal beats, and we select 2627 segments with PVC present and 356 segemnts with Atrial Premature Beat (APB) present. Experiments on both two dataset show quantitative evidence that introducing the segmentation of abnormal beats through an attention map, although unsupervised, can in fact benefit the arrhythmia classification performance as measured by accuracy, sensitivity, specificity, and area under Receiver Operating Characteristic (ROC) curve. At the same time, a grade study by experts qualitatively demonstrates our method's promising capability to segment abnormal beats among normal ones, which can provide useful insight into the classification result. Our code and dataset, which is the first for the challenging PVC differentiation problem, will be released to the public.

## 2 RELATED WORKS

**Multitask learning** There are many works devoted to training one deep learning models for multitasks rather than one specific task, like simultaneous segmentation and classification. (Yang et al., 2017) solves skin lesion segmentation and classification at the same time by utilizing similarities and differences across tasks. In the area of ECG signals, (Oh et al., 2019) modifies UNet to output the localization of r peaks and arrhythmia prediction simultaneously. What those two works have in common is that different tasks share certain layers in feature extraction. In contrast, our segmenter and classifier are independent models and there is no layer sharing between them. As can be seen in Figure 1, we use attention maps as a bridge connecting the two models. (Mehta et al., 2018) segments different types of issues in breast biopsy images with a UNet and apply a discriminative map generated by a subbranch of the UNet to the segmentation result as input to a MLP for diagnosis. However, their segmentation and classification tasks are not trained end-to-end. (Zhou et al., 2019) proposes a method for collaborative learning of disease grading and lesion segmentation. They first perform a traditional semantic segmentation task with a small portion of annotated labels, and then they jointly train the segmenter and classifier for fine-tuning with an attention mechanism, which is applied on the latent features in the classification model, different from our method. Another difference is that for most existing multitask learning works, labels for each task are necessary, i.e., all tasks are supervised. Our method, on the other hand, only requires the labels of one task (classification), leading to a joint supervised/unsupervised scheme.

**Attention mechanism** After firstly proposed for machine translation (Bahdanau et al., 2014), attention model became a prevalent concept in deep learning and leads to improved performance in various tasks in natural language processing and computer visions. (Vaswani et al., 2017) exploits self-attention in their encoder-decoder architecture to draw dependency between input and output sentences. (Wang et al., 2017) builds a very deep network with attention modules which generates attention-aware features for image classification and (Oktay et al., 2018) integrates attention gates into U-Net (Ronneberger et al., 2015) to highlight latent channels informative for segmentation task.

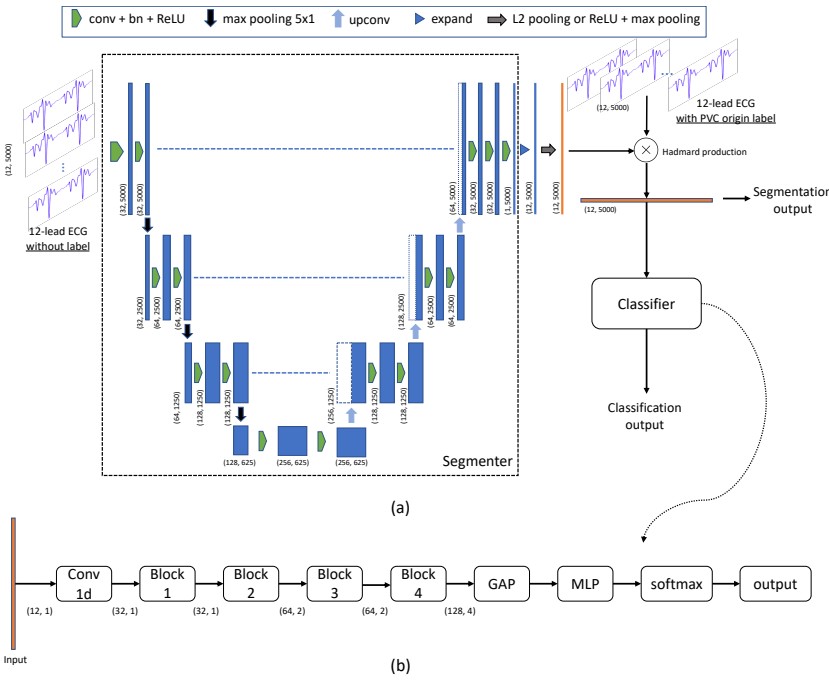

Figure 1: (a) Overview of our framework consisting of a segmenter and a classifier. The segmenter is a modified version of U-Net. The input of the classifier is the element-wise production of the original ECG segment and the attention map generated by the segmenter. The tuple alongside every cuboid represents (number of channels, length of data) for the feature maps. (b) The detailed architecture of the classifier. Following a 1D point-wise convolutional layer are four convolutional blocks, each containing two combinations of Conv + BN + ReLU. GAP stands for global average pooling while MLP stands for a two-layer perceptron. The elements in the tuple represent channel number and downsample ratio respectively.

When it comes to ECG, (Hong et al., 2019) proposes a multilevel knowledge guided attention network to discriminate Atrial fibrillation (AF) patients, making the learned models explainable at beat level, rhythm level, and frequency level, which is highly related to our work. Our method and theirs however are quite different in the way attention weights are derived and applied, as well as the output of attention network. First, in that work, the attention weights are obtained from the outputs of hidden layers, while ours are directly from the input. Second, domain knowledge about AF is needed to help the attention extraction, so the process is weakly supervised, while ours do not use any external information and is fully unsupervised. Third, their attention weights are applied to latent features in that work while ours are applied to the input for better interpretability. Finally, in that work, the input ECG segment is divided into equal-length segments in advance and the attention network output only indicates which segment contains the target arrhythmia. The quality highly depends on how the segment is divided, and it does not provide the exact locations of abnormal beats. On the other hand, our method directly locates the abnormal beats on the entire input ECG, offering potentially better interpretability and robustness.

## 3 METHOD

### 3.1 OVERVIEW OF THE FRAMEWORK

Here we briefly introduce the workflow of our joint learning frameworks for supervised classification and unsupervised segmentation. Firstly, in this work, we choose to model the input signal as a one-dimensional signal $D \in \mathbb{R}^{M \times N}$, where M is the number of leads and N is the length of the input ECG segment (number of samples over time). We then use a one-dimensional (1D) fully convolutional network called segmenter $S$ to output a feature map $L = S(D) \in \mathbb{R}^{M \times N}$, After that,

we apply a pooling layer to generate window-style element-wise attention $A \in \mathbb{R}^{M \times N}$, containing weights directly for every sample in the input ECG. The after-attention signal $X = A \odot D \in \mathbb{R}^{M \times N}$, where $\odot$ represents element-wise production, is then fed into a multi-layer CNN called classifier $C$, in which the outermost fully connected layer gives the prediction of the arrhythmia types. After training, the abnormal areas are highlighted in $X$, thus achieving the goal of segmenting abnormal beats from normal ones. Moreover, $x$, which indicates those beats that are highly associated with the differentiation task, also serves as an explanation for $C$'s decision. The architecture of our framework is illustrated in Fig 1.

## 3.2 SEGMENTER AND CLASSIFIER

In most existing works, the attention map is fused with the deep features in a neural network. However, for our specific purposes of enhancing intepretability of the classification results as well as unsupervised segmentation, the best result would be obtained by directly applying it to the input signal $D$. In order to generate attention weights of the same length as $D$, we choose to utilize U-Net (Ronneberger et al., 2015), a fully convolutional network highlighted by the skip connection on different stages. Encoding path extract features recursively and decoding path reconstruct the data as instructed by loss function. Note that the output of $S$ has only 1 channel and we expand it channel-wise so that it matches the channel dimension of the ECG signal and at the same time each channel gets the same attention. The reason is that the 12 leads are measured synchronously and the abnormal beats occur at the same time across all the leads.

Both recurrent neural networks (RNN) and CNN are candidate architectures for many arrhythmia classification works. RNN takes an ECG signal as sequential data and is good at dealing with the temporal relationship. CNN focuses on recognition of shapes and patterns in ECG, thus is less sensitive to the relative position of abnormal beats with respect to normal ones. Because abnormal beats may occur randomly among normal beats, we decide to use CNN as the backbone of our classifier. The detailed implementation of $C$ is shown in 1(b).

## 3.3 POOLING FOR WINDOW-STYLE ATTENTION

We do not use the output of the segmenter $L$ as the attention map directly but instead perform a pooling with large kernel size first. This is out of considerations for both interpretability and performance. Regarding interpretability, it is desirable that each abnormal beat is uniformly highlighted, i.e., the attention weights should be almost constant and smooth for all the samples within each abnormal beat. Regarding performance, it is desirable that the attention map $A$ does not distort the shape of abnormal beats after it is applied to the input $X$. Pooling layer is the easiest way to achieve this goal, functioning as a sliding window over multiple samples in an ECG signal for global information extraction. Max pooling outputs the same value around a local maximum, and average pooling reduces fluctuation by averaging over multiple samples. The kernel size cannot be too large, which may fuse sharp changes from neighboring areas and lead to the loss of local information. Therefore, deciding the proper pooling kernel size is essentially finding a balance between local information and global information preservation. Through experiments to be shown in Section 5.3, we find that setting the kernel size as nearly half the length of a normal beat yields the best balance between performance and interpretability. Padding of zeros on both sides of the segmenter output $L$ is implemented to keep the length of the resulting attention map $A$ after pooling to be the same as the input $X$.

Meanwhile, the polarization of QRS complex is a critical feature for ECG signal, while the commonly used pooling layers, like max pooling and average pooling, fail to control the sign of output, leading to differentiation performance degradation. Rectified linear unit (ReLu) $\sigma(l_{c,m}) = max(0, l_{c,m})$, where c and m denote channel number and spatial position in $L$ respectively, is usually performed as an activation function to add non-linearity to neural network for stronger representation ability. In this work, we can apply ReLU on $L$ before pooling so that the all the weights in $A$ generated by the following max pooling are positive. Alternatively, we replace average pooling with L2 norm pooling that takes square root of the L2 norm of input. In that case, ReLU is not needed. The two pooling implementations can be expressed as:

$$a_{c,m}{}^{max} = P^{MAX}(L)_{c,m} = max\left\{\sigma(l_{c,m}), \sigma(l_{c,m+1}), ..., \sigma(l_{c,m+k})\right\} \qquad (1)$$

$$a_{c,m}{}^{L2} = P^{L2}(L)_{c,m} = \sqrt{\sum_{i=m}^{m+k} l_{c,i}^2} \qquad (2)$$

where, $a_{c,m}$ is the $m$th data point in channel $c$ of $A$ and k is the kernel size for the pooling.

### 3.4 JOINT LEARNING

Compared to traditional segmentation network, our segmenter $S$ does not give a prediction on every point in the input ECG, instead we generate an attention map $A$ and distinguish heartbeats by the amplitude of weights in $A$. The segmentation result is reflected in the after-attention signal $X$. During training, with only annotation for the arrhythmia differentiation task, the segmentation task is actually unsupervised. Unlike clustering or mutual learning, the popular unsupervised segmentation methods, we train the segmenter and classifier in an end-to-end manner and the gradient of classification loss is backpropagated to $S$ for updating how the signal is segmented.

## 4 EXPERIMENTS

### 4.1 DATASET

Our experiments are conducted on two datasets from different sources. The first dataset is collected by a MAC5500 machine at a sample rate of $500Hz$. They are all from patients diagnosed with PVC. Further catheter ablation test is performed to confirm the origins of PVC, so all the labels are accurate. For every patient, experienced cardiologists exam the long ECG record and grabs a ECG segment that contains the PVC arrhythmia with fixed length of 5000 samples. In the dataset, there are totally 508 segments, including 135 cases of left ventricle (LV) and 373 cases of right ventricle (RV). Moreover, within left ventricle patients, 91 are left ventricle outflow origins (LVOT) and among right ventricle patients 332 are right ventricle outflow origins (RVOT). It is of clinical interest to classify between LV and RV, and between LVOT and RVOT, so we will explore both problems in our experiments. We will release this dataset to the public, which will be the first for the challenging problem of PVC differentiation.

The other dataset is derived from the public MIT-BIH Arrhythmia dataset (Moody & Mark, 2001), which includes 48 recordings of 47 patients, all sampled at 360 Hz. There are two leads for every records. All heart beats in those recordings are annotated by expert cardiologists and arrhythmia types include PVC, atrial premature beat (APB), left/right bundle branch block beat, etc. We preprocess those ECG signals into segment with standard 2000 samples. More specifically, we accumulatively add beat of interest to a segment until its length will exceeds 2000 if the next beat is added. Then we pad zeros for that segment to the target size. Among those segments, we focus on 356 segments with only normal beats and APB and 2627 segments with only normal and PVC beats.

As for preprocessing, we adopt a series of filters and remove the high frequency noise and baseline drift. Besides, we apply normalization to each lead independently so that the voltage ranges are all the same for the 12 leads.

### 4.2 EXPERIMENT SETTING

All our codes are based on the open source machine learning library PyTorch. Architecture details of our segmenter and classifier are shown in Figure 1. As for the classifier, inside each block are two serialized Conv + BN + ReLU combinations. The dimensions of weights in the two-layer perceptron are $128 \times 128$ and $128 \times 2$ respectively. We choose Adam algorithm (Kingma & Ba, 2014) as our optimizer with initial learning rate set to 0.00001. The training epoch is set to 120 as we observe lowest validation loss and highest accuracy can be achieved by then. The loss function for the classification is negative log likelihood loss and we add weights for different classes due to the imbalanced distribution of the dataset. We apply five-fold cross-validation with different classes evenly distributed between folds, and the average performance is reported.

We implement our method with L2 norm pooling as well as a combination of ReLu and max pooling at the output of the segmenter, as discussed in Section 3.3. All the hyperparameters remain the same

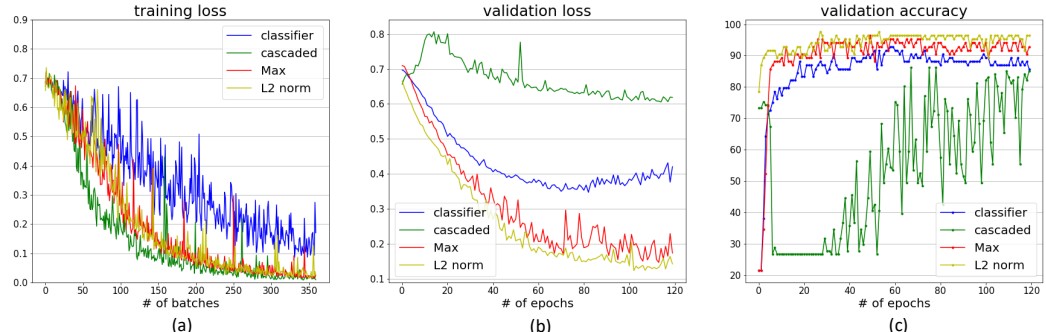

Figure 2: (a) Training loss curve, (b) validation loss curve and (c) validation accuracy curve for the classifier only method (Classifier), cascaded segmenter and classifier (Cascaded), our methods with ReLu and max pooling (Max), and with L2 norm pooling (L2 norm) on (LVOT, RVOT) task. There is a clear overfitting for the cascaded segmenter and classifier method.

for both datasets, except for the kernel size which is set to be linearly proportional to the sampling frequency as different sample frequencies result in different beat duration. The kernel size for PVC origin dataset is 200 and that for MIT-BIH dataset is 150. As none of the existing works addresses the problem of supervised arrhythmia classification with unsupervised abnormal beat segmentation, we also implement the following two baseline models for comparison: (1) **classifier only**: Keep the same classifier design, but without segmenter or any attention. (2) **cascaded segmenter and classifier**: Feed $A$ instead of $A \odot D$ into the classifier and train segmenter and classifier together. The resulting architecture will have the same number of layers as the original one in Figure 1.

## 5 RESULTS

| Task | Method | Accuracy% | Specificity% | Sensitivity% | AUC |
|------|--------|-----------|--------------|--------------|-----|
| (RV, LV) | Classifier only | $88.5 \pm 0.5$ | $93.5 \pm 0.9$ | $\mathbf{74.8 \pm 1.0}$ | $0.912 \pm 0.005$ |
| | Cascaded segmenter/classifier | $70.5 \pm 2.5$ | $78.8 \pm 5.1$ | $47.6 \pm 4.2$ | $0.728 \pm 0.030$ |
| | Ours (ReLu + max pooling) | $89.9 \pm 0.4$ | $96.4 \pm 1.0$ | $72.0 \pm 2.8$ | $0.918 \pm 0.014$ |
| | Ours (L2 norm pooling) | $\mathbf{90.2 \pm 0.6}$ | $\mathbf{96.5 \pm 0.4}$ | $72.7 \pm 2.6$ | $\mathbf{0.922 \pm 0.010}$ |
| (RVOT, LOVT) | Classifier only | $89.3 \pm 0.5$ | $94.6 \pm 0.8$ | $70.1 \pm 1.5$ | $0.878 \pm 0.019$ |
| | Cascaded segmenter/classifier | $70.2 \pm 1.9$ | $79.2 \pm 1.9$ | $37.4 \pm 11.6$ | $0.618 \pm 0.066$ |
| | Ours (ReLu + max pooling) | $89.6 \pm 0.3$ | $96.0 \pm 1.3$ | $70.0 \pm 0.6$ | $0.884 \pm 0.010$ |
| | Ours (L2 norm pooling) | $\mathbf{91.3 \pm 0.9}$ | $\mathbf{97.4 \pm 0.9}$ | $\mathbf{70.3 \pm 1.9}$ | $\mathbf{0.913 \pm 0.010}$ |
| (PVC, APB) | Classifier only | $98.1 \pm 0.4$ | $98.9 \pm 0.5$ | $94.9 \pm 1.3$ | $0.988 \pm 0.004$ |
| | Cascaded segmenter/classifier | $88.1 \pm 2.1$ | $96.3 \pm 4.4$ | $60.4 \pm 8.1$ | $0.928 \pm 0.050$ |
| | Ours (ReLu + max pooling) | $98.8 \pm 0.2$ | $99.4 \pm 1.0$ | $93.0 \pm 1.6$ | $0.990 \pm 0.007$ |
| | Ours (L2 norm pooling) | $\mathbf{98.8 \pm 0.1}$ | $\mathbf{99.6 \pm 0.2}$ | $\mathbf{93.4 \pm 1.3}$ | $\mathbf{0.993 \pm 0.004}$ |

Table 1: Comparison of different methods' performance on two tasks of PVC orgin differentiation and the arrhythmia classification task on MIT-BIH. Our method with L2 norm pooling almost always attains the best scores.

### 5.1 COMPARISON OF PVC DIFFERENTIATION PERFORMANCE

The metrics we select for comparison include overall accuracy, specificity, sensitivity, and AUC (area under curve) of ROC (Receiver Operating Characteristic) curve. We evaluate the performance of all the methods on two tasks: differentiating PVC originating in LV and RV ((RV,LV) task), as well as PVC originating in LVOT and RVOT ((RVOT, LOVT) task). For (RV, LV) task, the specificity and sensitivity are calculated with regards to RV, for (RVOT, LOVT), it is with regards to RVOT, and it is PVC for (PVC, APB) task. For all tasks, we calculate the AUC of ROC curve for each class and record the average value.

Table 1 lists the results for three baseline methods and our methods with two different pooling. A few meaningful observations can be made out of the table. Firstly, in general our methods show higher

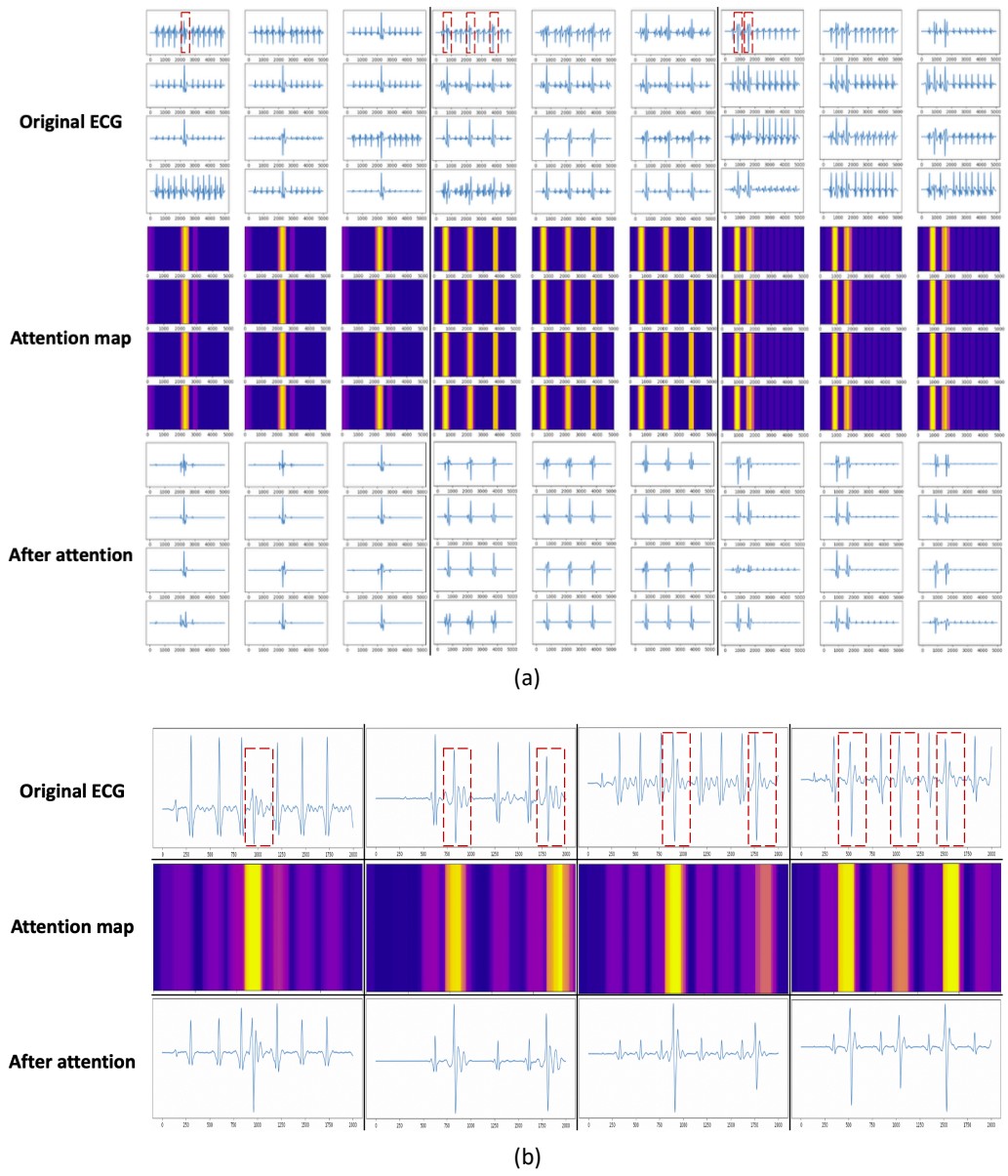

Figure 3: Visual examples of the segmentation results on (a) our dataset and (b) MIT-BIH Arrhythmia dataset. The columns represent different cases. For (a), in the original ECG, the abnormal beats occur simultaneously in all the 12 leads and are only marked on the first one. Regarding the heat map of the attention, the warmer an area is, the larger the attention weight is.

score in almost all benchmarks including accuracy, specificity, sensitivity and AUC, than the baseline methods. This proves that our attention mechanism indeed improves the classifier's capability of PVC origin classification. Secondly, using L2 norm pooling has better performance than using the combination of ReLU and max pooling, which implies the limitation of ReLU which may lose information in negative values. In contrast, L2 norm pooling preserves the negative information. Finally, the cascaded segmenter and classifier method, which has the same number of layers as our methods, has poor performance. This confirms that the better classification performance of our methods actually comes from the attention mechanism instead of deeper architecture. Actually, from Figure 2 we can see that there is a large gap between the training loss and the validation loss for the cascaded segmenter and classifier method after several epochs, suggesting apparent overfitting.

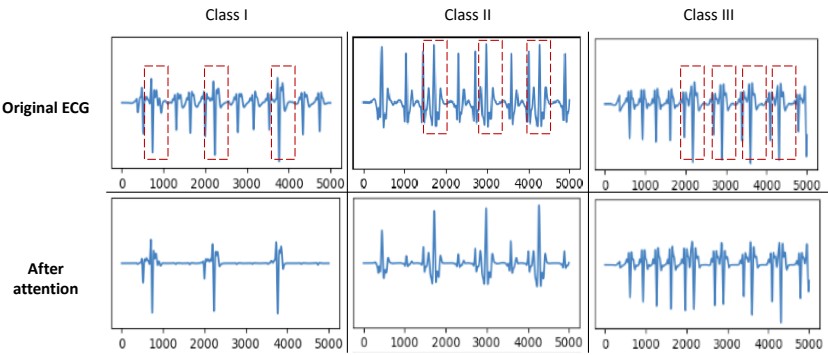

Figure 4: Illustration of the three classes in grade study. In class I, all abnormal beats are identified and all normal beats are removed. In Class II, all abnormal beats are detected, though some normal beats are also included. In Class III, no significant difference can be found between abnormal and normal beats.

Moreover, the benefits of adding a segmenter to the MIT-BIH dataset seems not large, but it is due to that distinguishing between PVC and APB is much less challenging than differentiating various origins of PVC. Also, the baseline already achieves good performance.

## 5.2 EVALUATION OF SEGMENTATION AND INTERPRETABILITY

Three visual examples of the segmentation results are shown in Figure 3. From the figure we can see that after applying the attention map to the original ECG signal, the location of the abnormal beats can be easily identified.

We design an independent and blind grade study by an experienced cardiologist to qualitatively evaluate our segmenter's ability to detect abnormal beats for the (RVOT, LVOT) task. In general, qualitative evaluation is widely used in attention mechanism related works due to simplicity and visualization (Hu, 2019). It is also most suitable to judge the intepretability perspective of the results.

Focusing on the after-attention signal $X$, we categorize the segmentation result into three classes by the contrast between abnormal beats and normal beats, as shown in Figure 4. Class I: all normal beats are eliminated, and all abnormal beats are kept. Best interpretability is attained. Class II: some normal beats still remain, and all abnormal beats are kept. In this case, interpretability is reduced but all abnormal beats can still be identified in $X$. Class III: there is no significant difference between abnormal beats and normal beats. There is little interpretability.

We randomly select the after-attention signals of 100 ECG segments and the blind grade study result shows that the number of cases in the three classes are 50:27:23 (Class I: Class II: Class III), which implies superior performance of our segmenter. In practice, these segmentation results provide cardiologists quick understanding of why the prediction is made.

## 5.3 INFLUENCE OF KERNEL SIZE

Fig 5 shows the comparison of classification performance and segmentation result of our framework with different kernel sizes in the L2 norm pooling layer at the output of the segmenter regarding the (RVOT, LVOT) task. We can see apparent degradation of accuracy, specificity, and AUC when the kernel size is increased to 300 while the performance difference between kernel size 100 and 200 is minor. Regarding the segmentation result, additional grade studies are conducted on kernel size 100 and kernel size 300. The ratios between the number of cases in the three classes when the kernel size is 200 is close to those when kernel size is 300, showing comparable ability of abnormal beats detection. There is a much higher number of class III cases for kernel size 100, suggesting that too small kernels may suffer from poor segmentation and low intrepretability in accordance with the

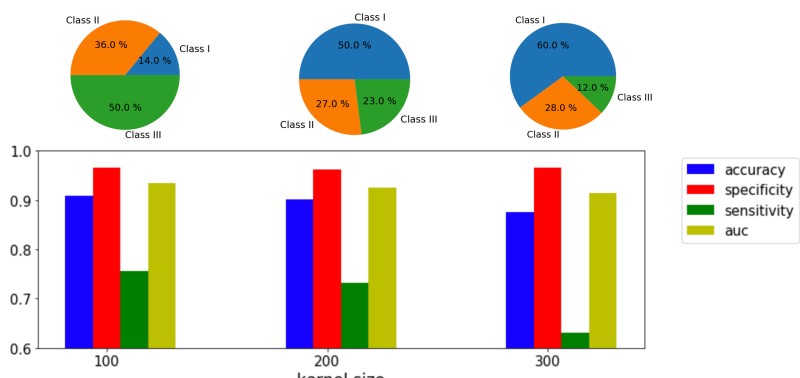

Figure 5: Comparison of classification performance and grade study result for different kernel size with respect to differentiating RVOT and LVOT

analysis in Section 3.3. After weighing interpretbility and performance, we choose the kernel size of 200.

## 6    CONCLUSION AND DISCUSSION

In this paper, we propose a novel framework combining unsupervised abnormal beats segmentation and supervised arrhythmia differentiation. The key to the multitask learning is applying an attention map generated by a segmenter directly to input data before the classification task. In addition, we perform a large-kernel pooling layer to constrain the shape of attention map for better performance and easier interpretability. We use premature ventricular contraction differentiating, one of the most challenging problems in arrhythmia classification as a case study to evaluate effectiveness of our method. On one hand, experiment result demonstrates better accuracy with the help of attention map. On the other hand, we observe obvious discrimination between abnormal beats and normal beats from after-attention signal. Indicating enhanced interpretability in clinical practice.

In the future, we expect to extend our method to high-dimension data such as images and videos. In our opinion, the difficulties in applying our framework on 2-D /3-D data are more complicated background information and the need for fine-grained constraint on the attention map shape. When doing arrhythmia classification, the "background" in ECG signal is just normal beats. As for image classification, the "background" can be more complex, like birds flying among flowers and cars driving through streets, etc. Learning the difference between target objects and environment may be harder for the segmenter if without labels. On the other hand, the target objects can have higher variations in terms of size, shape and texture, even within the same class, which requires a more elaborate design of constraints on the attention map shape.

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
