# OpenReview forum: "Attention Based Joint Learning for Supervised Electrocardiogram Arrhythmia Differentiation with Unsupervised Abnormal Beat Segmentation"
_ICLR.cc/2021/Conference — Reject_

### Official Review · AnonReviewer2 · 2020-10-26
**Interesting paper, but the topic is too narrow; related image segmentation works needed**

**Rating:** 4
**Confidence:** 4

**Review:**

This paper proposes a deep neural network for Premature Ventricular Contraction (PVC) differentiation and segmentation from electrocardiogram (ECG) signals. The network is jointly trained as a segmenter and a classifier with a multitask learning manner. Differentiation is achieved by the classifier, and segmentation is achieved by pooling for window-style attention from segmenter’s output. Quantitative experiments show better performance than baselines on differentiation tasks. Qualitative experiments show the effectiveness of segmentation tasks.

The results look interesting, and it might have a broader impact on practical usage for AI models in the clinical environment. However, my concerns are:

1) The topic seems too narrow for the computer science community. More likely a paper of the biomedical engineering community or computing cardiology community. The proposed method also lacks in-depth technical/theoretical analysis; thus the paper novelty is limited.

2) The related works include multitask learning and attention mechanisms. But (image) segmentation works are also worth (or even more) investigating. Just a simple modification of image segmentation neural networks (such as Conv2D -> Conv1D) can make them suitable for ECG segmentation tasks.

3) For the evaluation of segmentation, only several cases of qualitative evaluations are not convincing. At least, a comprehensive user study by a community of cardiologists is needed.

Some questions:

- Could you provide more details about data preprocessing? Which filters do you use? What are the cut-off frequencies for high-pass filter and low-pass filter?

- In figure 3, are there duplicate attention maps in every column?

---

> ### Author Response · Authors · 2020-11-16
> **Response to some comments and questions**
>
> 1, Regarding the comment “The topic seems too narrow for the computer science community.”
>
> We add extra experiment on other public ECG dataset to demonstrate the generalization of our methods on ECG classification problem. The accuracy and AUC-ROC increases by 0.006 and 0.002 respectively with a segmenter added to the classifier. Meanwhile, we also observe a promising segmentation result.
>
> 2, Regarding the comment “But (image) segmentation works are also worth (or even more) investigating.”
>
> It’s worth noticing that our work focus on unsupervised segmentation. It’s true that the modification of image segmentation models can fit in ECG segmentation, and this is actually how some works concerning ECG segmentation do.  However, they all need annotations, as we mentioned in paragraph 2 in section 1, while in our work we focus on unsupervised segmentation. We notice that there are emerging works concerning unsupervised image segmentation methods very recently. Yet due to the very different nature between nature images and ECG signals, these unsupervised methods can hardly be directly applied to ECG segmentation.
>
> 3, Response to the concern about the evaluation of segmentation
>
> For evaluation of segmentation results, the example we gave in Fig 4 is the illustration of the three classes in the grade study. Actually, we asked independent expert cardiologists to do blind grade study on the segmentation result (100 ECG segments) and the result is shown in Fig 5. We think it would be too much work for a conference paper to conduct multi-site study (and it is very rare too).
>
> 4, More details about data preprocessing
>
> For data prepcocessing, we use butter filter to build a low-pass filter with threshold frequency of 60. What’s more, we apply a low pass FIR filter to remove the baseline drift and the cutoff frequency is 4.
>
> 5, Response to “In figure 3, are there duplicate attention maps in every column?”
>
> Yes, there are. We enforce the output of U-Net to have only one channel and duplicate it into 12 copies so that the attention maps for 12 leads are exactly the same. This is because the arrhythmia occurs synchronously for the 12 leads.

---

### Official Review · AnonReviewer4 · 2020-10-28
**This paper presents an semi-supervised approach for ECG segmentation and PVC classification. The application is well motivated. I have some concerns about the experimental evaluation and novelty described below.  I think it has the makings of a promising paper but would like to see responses to these questions.**

**Rating:** 5
**Confidence:** 4

**Review:**

This paper presents a method for segmentation and classification of ECG data applied to the task to segmenting and detecting Premature Ventricular Contractions (PVC). The taks is semi-supervised, in the sense that segmentation labels are not required by labels for the PVC events (classification) are used.
The authors motivate this application quite well and detecting abnormalities in ECG signals is an important task of clinical relevance.  I can understand why segmentation labels may be very laborious to collect and unsupervised methods would be desirable.

The proposed approach builds upon U-Net and introduces some task specific changes.  However, I would argue that this is primarily an application paper. I don't mean that as a criticism necessarily, I think that strong and well motivated applications of machine learning are important and informative. However, it would be helpful if the authors could discuss more about how their approach might generalize to other tasks, both the detection of other types of arrythmias and other temporal segmentation and classification tasks.

My main comments regarding the paper are around the experimental evalutation.  The authors highlight that there are some published baselines for this task or at least similar related works (e.g., Moskalenko et al. (2019); Oh et al. (2019)) and/or the authors could have applied classification on top of features extracted using Pan-Tompkins - but that would be a more crude baseline.  While I recognize that these approaches might not enable unsuperivsed segmentation and so direct comparisons on that might be hard with the full approach they propose.  It might be possible to present a comparison of classification metrics on their own. Perhaps I am misunderstanding but it doesn't seem as though Table 1 includes such a comparison, rather the baselines are different from the previous published methods - is that correct?  I would almost describe Table 1 as ablation results rather than a comparison with other published baselines. I'd like to know the author's response to that and if Table 1 does show these results perhaps linking the rows to the previous approaches might be helpful?  Or justifying why it isn't appropriate to show these comparisons. I don't say this just because the authors should show better numbers, but rather to ground the chose baselines in the context of previous work in this space.

Building from the previous point. I think this paper would be an excellent case for for showing transfer learning results, it seems to me that PhysioNet provides a large amount of available data for ECG classification.  A couple of question I'd like to hear the authors responses to:
1) Why did they not do any experiments on these public datasets?  Is there a reason they are not appropriate?  Do they not have the right labels, are they not large enough, do you need full 12 lead recordings (I am not sure if they are avaiable on PhysioNet datasets - but I imagine so.)
2) Even if training your method on your dataset is preferable, it would seem natural to test it on a set from PhysioNet, perhaps even with a different type of arrythmia, to see how much performance degrades? This I think would be most informative, both showing segmentation and classification results.

Fig. 3 is a nice illustration, but it is quite difficult to read.  I might suggest reorganizing it.  I am not sure showing multiple leads is necessary and maybe limiting to two columns might help.  I'd encourage the authors to leverage supplementary material to show more examples as I do think these help.

Finally, physiological signals are notorious for having large individual variation.  I'd be interested to have the authors discuss more about this. I couldn't find the information about how the train/val/test splits were organized and whether this was person independent etc.  The following sentence in Section 4.2 "We apply five-fold cross-validation with different classes evenly distributed between folds, and the average performance is reported" doesn't seem to mention that.  Knowing more about the splits would be very helpful.  This is perhaps another reason that performing experiments on at least one PhysioNet dataset would be helpful as the train, val, test splits could be released.  But I acknowledge that the authors say they will release their data which is good.

---

> ### Author Response · Authors · 2020-11-16
> **Response to some comments and questions**
>
> 1, Response to the concern about the experimental evaluation
>
> Actually, Moskalenko et al. (2019); Oh et al. (2019) deals with ECG segmentation problem and the Pan-Tompkins algorithm is used for finding QRS complexity’s position in an ECG signal, and none of them can be used for ECG classification. Table 1 lists the comparison of classification results of different methods, which are more related to the works mentioned in the first paragraph of section 1. The commonly used models for ECG classification include CNN and CRNN. Different works make problem-specific modification to CNN or CRNN for the target problem/dataset and there does not exist a state-of-the-art approach for the PVC differentiation problem. The “classifier only” in Table 1 stands for the CNN baseline similar to the state-of-the-art in many other ECG classification problems, while we get poor classification performance with CRNN so that we didn’t present it. Note that Hong et al., (2019) actually uses a similar baseline for comparison without referring to a specific previous work.
>
> 2, Response to the two questions about PhysioNet
>
> Regarding PhysioNet, we did run experiments on a dataset derived from public MIT-BIH dataset to classify ECG segments with atrial premature beat (APB) and ones with premature ventricular contraction (PVC). For this task, the baseline method only using CNN models achieves almost 99% accuracy, so the necessity for adding a segmenter is minor. Still, we do observe improvement of classification performance with our methods and a promising segmentation result. We will add it to the new version of our paper. On the other hand, the problem of differentiating subclass of PVC is more challenging with low classification accuracy, so there is a large room for improvement.  As for transfer learning, each ECG classification problem is quite unique in terms of features and PVC differentiation is among the most difficult ones. Therefore transfer learning does not work well.
>
> 3, Regarding how the train/val/test splits are organized
> It’s a good point to mention physiological signals’ large individual variation. Actually, for each patient, there are at most three segments and in many cases one patient has only one segment. When doing k-fold validation, we split patients not segments into k folds.

---

### Official Review · AnonReviewer3 · 2020-10-28
**CNN-based approach for segmentation and classification of ECG signals which is quite ad-hoc and limited novelty**

**Rating:** 6
**Confidence:** 4

**Review:**

The paper proposes a framework for the classification of arrhythmias in electrocardiogram (ECG) data. The proposed approach performs segmentation and classification of the ECG signal. The segmenter performs segmentation of the signal (also called attention map) even though the term segmentation is not quite correct. This attention-modulated signal is then classified to identify the origin of Premature Ventricular Contraction (PVC).  The proposed approach is evaluated on a dataset from a single machine consisting of 508 segments (I am not sure what “segments” means in this context). The results seem ok, but it is not clear to me what level of performance is required in order to achieve a similar level of performance as an expert.

Main concern is that the proposed approach seems rather ad-hoc: The combination of segmentation (or attention) and classification in a joint fashion seems hardly new and while the results obtained are good, there is no systematic evaluation how the method compares to other state-of-the-art ECG classification methods. Another problem is that the writing in the paper is not always clear and it is often unclear what exactly the authors are doing. As a result, it is quite difficult to exactly assess what the authors have done or what they mean.

Detailed comments:

• What is the output of the classifier? Is this a binary label? Or a multi-class label?
• The authors write “… the output of S has only 1 channel and we expand it channel-wise so that it matches the channel dimension of the ECG signal …” – What exactly is meant here? In Fig 1 it seems that the segmentation output has naturally 12 channels? Should the segmentation be identical for all channels?
• “We do not use the output of the segmenter L as the attention map directly but instead perform a pooling with large kernel size first” – Why is this done? What does “large kernel” mean?
• Where is the attention map in Fig. 1?
• How are the Premature Ventricular Contraction (PVC) origin labels defined? Is that a single time point (per channel or common for all channels) or a time window?

---

> ### Author Response · Authors · 2020-11-16
> **Response to some comments and questions**
>
> Thank you for your advices, we will edit our paper trying to make our writing more understandable.
>
> 1, Regarding the comment “the proposed approach seems rather ad-hoc"
>
> I agree that combining segmentation and classification is not a novel invention. However, our contribution is a new way of unsupervised learning through the assistance of supervised learning on a related but different task. By combining unsupervised segmentation and supervised classification in the form of attention maps directly on input images, which provides some explainability to the classification task.
>
> 2, Regarding the comment “there is no systematic evaluation how the method compares to other state-of-the-art ECG classification methods”
>
> As for comparison to other state-of-the-art ECG classification methods, when focusing on classifying a segment of ECG segment, there are two types of widely used methods, which are CNN and CRNN (combination of CNN and RNN). In Table 1, the “classifier only” represents the CNN baseline. We also tried CRNN method but the accuracy is only around 70% due to the challenging nature of the PVC differentiation problem as explained in Section 1. Hence we chose not to present the result.
>
> Response to detailed comments:
>
> i. What is the output of the classifier? Is this a binary label? Or a multi-class label?
>
> The labels are binary. Please kindly refer to Section 5.1, where we have made it clear “We evaluate the performance of all methods on two tasks: differentiating PVC originating in LV and RV, as well as PVC originating in LVOT and RVOT”.
>
> ii. “… the output of S has only 1 channel and we expand it channel-wise so that it matches the channel dimension of the ECG signal …” – What exactly is meant here? In Fig 1 it seems that the segmentation output has naturally 12 channels? Should the segmentation be identical for all channels?
>
> The segmentation result should be identical for all channels since the abnormality occurs at the same time for all 12 leads. To enforce that, we set the output of the segmenter to have only one channel and duplicate it 12 times, which is called “expand” in the paper. After that, we apply a pooling layer. In Fig 1, the after-pooling attention map is not the output of segmenter s. We should make it clear in Fig-1 that the segmenter should not contain the two-step postprocessing afterwards.
>
> iii. “We do not use the output of the segmenter L as the attention map directly but instead perform a pooling with large kernel size first” – Why is this done? What does “large kernel” mean?
>
> We explain it shortly afterwards “out of consideration for both interpretability and performance”. We would like to generate a window-like attention map so that the abnormality area is uniformly highlighted in contrast to normal beats. Besides, if the attention weight varies in the “window”, the shape of abnormal beats would be distorted. As for “large kernel”, traditional 3*3, 5*5 max pooling layers’ kernel size is 3 and 5. Global max pooling’s kernel size is the shape of input signal. In our case, the kernel size is almost half of the beat length (e.g. 200). Compared to traditional 3*3 max pooling, our pooling has pretty “large” kernel.
>
> iv. Where is the attention map in Fig. 1?
>
> The attention map is marked by the orange line segment, the input to the Hadmard production.
>
> v. How are the Premature Ventricular Contraction (PVC) origin labels defined? Is that a single time point (per channel or common for all channels) or a time window?
>
> The PVC origin label is segment-wise, which means for each segment (12-leads) there is only one label denoting whether there are LV or RV (LVOT or RVOT) beats.

---

### Official Review · AnonReviewer1 · 2020-10-29
**Empirical evidence has some loopholes**

**Rating:** 5
**Confidence:** 4

**Review:**

This manuscript contributes a neural architecture to classify arrhythmia type from ECG data. The signal treated as 1D, and the architecture performs joint segmentation-classification detecting the abnormal beats and then classifying them as a function of their origine. It uses U-nets for segmentation and, for classification CNN and one fully-connected layer. The unet segmentation generates weights that are considered as an attention map and multipled with the original time series after pooling on a window (which amounts to smoothing).

Compared to the prior art, the central contribution put forward is the addition of the segmentation component of the architecture.

The work is light on theory and the contribution mostly resides on the empirical improvement. However, the evidence for this improvement is not rock solid, as it is shown on a single dataset, which has a rather small sample size. Also, I fear that hyper-parameters are not set fully independent of the final error measure.

How are hyper-parameters (such as learning rate or architecture parameters) chosen? Given the procedure exposed in section 5.2, it seems to me that some of the architecture parameters (kernel size) where not chosen independently of the test set. Such choice will incur a positive bias with regards to the actual expected generalization error.

With n=500 and an accuracy of 90%, the p=.05 confidence interval of a binomial model is 5%. Hence, the improvements observed by adding the segmentation on top of the classifier do not seem really significant.

---

> ### Author Response · Authors · 2020-11-16
> **Response to some comments and questions**
>
> 1, Regarding the comment “The work is light on theory and the contribution mostly resides on the empirical improvement”
>
> Our work’s contribution in theory is providing a new way of unsupervised learning through the assistance of supervised learning on a related but different task. In this particular problem, it is done by backpropagating the supervised classification loss to the unsupervised segmenter. Besides, we propose an explicit attention approach directly on the input signal, different from those in the literature which apply attention on intermediate features, for better explainability of the results. We further discuss why pooling is important for the attention map.
>
> 2, Regarding the comment “the evidence for this improvement is not rock solid, as it is shown on a single dataset, which has a rather small sample size”
>
> For more generalized conclusion, we have added the comparison results between our method and the baseline on a public dataset MIT-BIH to the new vision of our paper in section 4. The accuracy and AUC-ROC increases by 0.007 and 0.005 respectively with a segmenter added to the classifier. The dataset has total samples of almost 3000 and the baseline already reaches 0.98 (accuracy) and0.99 (AUC-ROC), so we think the improvement is acceptable. Meanwhile, we also observe a promising segmentation result.
>
> 3, Response to the question about hyperparameter selection
>
> In terms of hyperparameters, both learning rate and architecture parameters are fixed independent of the dataset (we used the same setting in the new experiment on the public dataset). The kernel size linearly depends on the sample frequency of the dataset, which is natural. For different datasets, we can normalize the sampling frequency to use the same kernel size, as has been shown in the newly added experiments on the public dataset.
>
> 4, Response to the "not really significant improvement”
>
> Lastly, thanks for providing a new perspective on evaluating the solidity of our result. Actually, when n=500 and assume the true accuracy is 90%, the confidence interval with 95% confidence is +/- 2.62%. I can understand your concern that the margin between the accuracy of our method and baseline is smaller than the 95% confidence interval. However, the hypothesis behind this calculation is that test results for different samples are independent, which is not true in our case. This could lead to smaller deviation. Besides, it’s hard to attain 5% accuracy increase anyway when the baseline accuracy already reaches 90%. On the other hand, this confidence interval theory only applies to accuracy not for specificity, sensitivity, and AUC-ROC. Therefore, by showing the comparison of these metrics, we can also conclude performance improvement with our method.

---

### Decision · Program_Chairs · 2021-01-07
**Final Decision**

**Decision:**

Reject

**Comment:**

This paper received 4 reviews with mixed initial ratings: 5, 6, 4, 4. The main concerns of R1, R4 and R2, who gave unfavorable scores, included: insufficient evaluation (lack of experiments on public datasets, small sample size), an ad-hoc nature and overall limited novelty of the method, a number of issues with the presentation. In response to that, the authors submitted a new revision and provided detailed answers to each of the reviews separately. After having read the rebuttals, the reviewers (including R3, who initially gave a positive rating) felt that this work overall lacks methodological novelty and does not meet the bar for ICLR.
As a result, the final recommendation is to reject.